# Promising Strategies for Transdermal Delivery of Arthritis Drugs: Microneedle Systems

**DOI:** 10.3390/pharmaceutics14081736

**Published:** 2022-08-19

**Authors:** Jitong Wang, Jia Zeng, Zhidan Liu, Qin Zhou, Xin Wang, Fan Zhao, Yu Zhang, Jiamiao Wang, Minchen Liu, Ruofei Du

**Affiliations:** 1Engineering Research Center of Modern Preparation Technology of TCM of Ministry of Education, Shanghai University of Traditional Chinese Medicine, Shanghai 201203, China; 2NHC Key Lab of Reproduction Regulation, Shanghai Institute for Biomedical and Pharmaceutical Technologies, Shanghai Engineering Research Center of Reproductive Health Drug and Devices, Shanghai 200032, China; 3Department of Rehabilitation, Baoshan Hospital of Integrated Traditional Chinese Medicine and Western Medicine, Shanghai 201999, China; 4School of Pharmacy, Shanghai University of Traditional Chinese Medicine, Shanghai 201203, China

**Keywords:** arthritis, microneedle, transdermal route, drug delivery

## Abstract

Arthritis is a general term for various types of inflammatory joint diseases. The most common clinical conditions are mainly represented by rheumatoid arthritis and osteoarthritis, which affect more than 4% of people worldwide and seriously limit their mobility. Arthritis medication generally requires long-term application, while conventional administrations by oral delivery or injections may cause gastrointestinal side effects and are inconvenient for patients during long-term application. Emerging microneedle (MN) technology in recent years has created new avenues of transdermal delivery for arthritis drugs due to its advantages of painless skin perforation and efficient local delivery. This review summarizes various types of arthritis and current therapeutic agents. The current development of MNs in the delivery of arthritis drugs is highlighted, demonstrating their capabilities in achieving different drug release profiles through different self-enhancement methods or the incorporation of nanocarriers. Furthermore, the challenges of translating MNs from laboratory studies to the clinical practice and the marketplace are discussed. This promising technology provides a new approach to the current drug delivery paradigm in treating arthritis in transdermal delivery.

## 1. Introduction

Arthritis is one of the important diseases affecting contemporary human health, and generally refers to inflammatory diseases that occur in human joints and surrounding tissues [1]. The clinical manifestations are mostly joint pain, swelling, deformity, and dysfunction, resulting in stiffness and deformation of the patient’s joints and progressive loss of self-mobility, which seriously affect the quality of patients’ life. The progression of inflammation and cartilage destruction, leading to joint disability in severe cases, has made arthritis one of the most important causes of disability. [2]. According to the pathogenesis, arthritis can be divided into several specific types. Common clinical arthritis includes rheumatoid arthritis (RA), osteoarthritis (OA), gouty arthritis (GA), juvenile idiopathic arthritis (JIA), and psoriatic arthritis (PSA). It has been estimated that 10 million Britons and 52.5 million Americans were diagnosed with arthritis, according to statistics published in 2020. The prevalence of arthritis over the age of 20 years was 24.7%, with OA accounting for 9.7%, RA for 4.2%, other arthritis for 2.8%, and unknown type  for  8.0% [3,4].

In the early stage of arthritis, drug treatment is the mainstay. To inhibit the development of inflammation and reduce pain, non-steroidal anti-inflammatory drugs (NSAIDs), glucocorticoids (GCs), disease-modifying antirheumatic drugs (DMARDs), immunotherapy drugs (ASOs), cytokine inhibitors (TNF-α inhibitors, nuclear factor-κB inhibitors, etc.), and cytokines are typically used [5]. Although drug therapy is able to ameliorate the disease, arthritis treatment medications require a long-term application, and the side effects associated with both oral delivery and injections cannot be ignored [6,7,8]. Transdermal drug delivery has emerged as a new option for the treatment of arthritis, with benefits such as fast local action, bypassing the first-pass effect, avoiding gastrointestinal reactions, and painless drug delivery, compared to the limitations of traditional drug delivery methods such as lack of local orientation, gastrointestinal reactions, and poor patient compliance [9].

However, the skin is a powerful biological barrier against infectious viruses and harmful substances [10]. In particular, transdermal drug delivery to the joints involves not only the skin barrier but also the joint capsule barrier, making it challenging to transfer drugs into the joint cavity efficiently by typical transdermal routes [11].

Microneedles (MNs) are a promising transdermal delivery approach that physically penetrates the skin and is more efficient in delivering drugs; they have gradually gained attention in the field of transdermal delivery [12]. Specifically, they are a needle-like drug delivery system with a needle length of 25–1000 μm and a pointer size in the micron range. MNs penetrate the stratum corneum (SC) to form microchannels that cross the barrier layer and carry drugs to the upper dermis and distribute them to the body circulation to produce systemic pharmacological effects. The length of the MNs can be increased to promote vertical diffusion of the drug, thus allowing the drug to accumulate locally, resulting in enhanced local pharmacological effects [13]. Moreover, since the length of the MNs is usually less than the thickness of the epidermis, it cannot reach the nerve endings and blood vessels in the dermis and causes no severe discomfort, which improves the patient’s compliance during application [12]. Furthermore, MNs have been shown to carry not only small molecules but also macromolecules such as proteins and peptides, offering additional choices for treating arthritis when compared to the standard transdermal medication delivery [14]. With the current developments in science and technology, MNs can be delivered not only in a controlled release but also in a targeted manner [15]. These technologies are beneficial to the transdermal drug delivery process in arthritis, reducing the number of doses and making it easier for patients to use; they maintain stable drug concentrations in blood, tissues, and specific targets, allowing the drug to work consistently [16,17]. MNs can also be loaded with nanoparticles to improve drug stability, solubility, dissolution, and bioavailability, as well as aid pharmaceuticals in reaching the target area more quickly [18]. A US patent filed by Russell Frederick Ross in 2010 described a MN transdermal device for the delivery of RA therapeutic drugs [19]. The device included MNs made of various materials and could deliver small molecules, macromolecules, and protein therapeutic agents, which demonstrated the feasibility of MNs in the delivery of RA therapeutic drugs. Until now, very few review articles on the treatment of arthritis with MNs have been published. 

In this paper, we further review the application of MNs delivering drugs for the treatment of various types of arthritis including RA, OA, GA, and JIA. The advantages and disadvantages of MN-based therapies for arthritis are further discussed, which will provide a strategy for designing and developing new MNs for arthritis in the future.

## 2. Types of Arthritis

Arthritis is an umbrella term that encompasses more than 100 joint pain or disease conditions for which there is not yet a uniform classification standard [20]. However, several of the most frequent types of arthritis have raised great clinical interest in developing new therapies, including RA, OA, GA, and JIA. Furthermore, research on the use of MNs is also on the rise. Features of these types of arthritis are shown in Table 1.

### 2.1. Rheumatoid Arthritis

RA is an autoimmune illness that is chronic, symmetrical, and inflammatory [21]. The point prevalence of RA is 0.51% worldwide [22]. It damages the bones, cartilage of joints, tendons, and ligaments, resulting in joint deformities and bone erosion. It initially affects small joints, then large joints, and finally other organs of the body [23]. The pathogenesis of RA is complex, mainly related to synovitis caused by immune system diseases, which leads to the activation of osteoclasts, then causes bone destruction and para-articular osteoporosis. The autoimmune response begins with self-antigen presentation by antigen-presenting cells, such as dendritic cells and macrophages, which promotes the expression of inflammatory factors including tumor necrosis factor (TNF) and interleukin 6(IL-6), and further activates synovitis and osteoclasts. The activated synovial cells express nuclear factor-κB receptor activator ligand (RANKL) to produce proteases that destroy bone and cartilage [24,25,26].

### 2.2. Osteoarthritis

OA is the most common type of arthritis in the world. It mainly occurs in the elderly and is caused by degenerative lesions with articular cartilage damage [27]. OA affects approximately 240 million people worldwide [28]. Its pathogenesis is not completely clear. At present, OA is known to be a complex process involving inflammatory and metabolic variables. Systemic inflammation plays a major role in the etiology of OA by activating synovitis [2]. One mechanism is that cartilage degradation causes a foreign body reaction in synovial cells, which leads to inflammation. Furthermore, the innate immune system is involved in the pathophysiology of OA, with physical stimulation or inflammatory stimuli causing it to produce an inflammatory response [29]. OA causes the generation of inflammatory cytokines including interleukin (IL)-1, IL-6, matrix metalloproteinase (MMP), TNF, and matrix-degrading enzymes such as collagenases, all of which will contribute to cartilage destruction.

### 2.3. Gouty Arthritis

According to a systematic investigation of the Global Burden of Disease Study, GA is a self-inflammatory disease that often occurs in men over 40 years old [30]. The global age-standardized prevalence rate of GA was 0.52% in 2017 [31]. It is caused by hyperuricemia and the deposition of monosodium urate (MSU) crystals in the joint capsule, bursa, cartilage, bone, and other tissues, which causes lesions and inflammatory reactions [31]. Macrophage uptake of MSU crystals activates NLRP3 inflammatory vesicles, a signal that leads to a cascade of inflammatory responses characterized by redness, swelling, heat, and discomfort in the joints [32]. IL-1β plays a central role in crystal-induced inflammation in GA, which provides a new idea for the drug treatment of GA [33].

### 2.4. Other Arthritis

Apart from above types of arthritis, which have attracted much attention in the clinical environment, other types of arthritis, such as JIA, PSA, ankylosing spondylitis (AS), and reactive arthritis (ReA), also affect the health of a group of people.

JIA is a broad term for all types of chronic childhood arthritis. It is defined as arthritis that occurs in children under 16 and lasts for at least 6 weeks for unknown causes [34]. JIA is divided into many heterogeneous subtypes, including oligoarticular JIA, polyarticular JIA, RF-negative, polyarticular JIA, RF-positive, enthesitis-related JIA, psoriatic JIA, systemic JIA (sJIA), and undifferentiated JIA [35]. JIA not only affects joints, but also affects other organs throughout the body, resulting in disability or death.

PSA is an inflammatory joint disease associated with psoriasis, embodied by psoriatic rash accompanied by joint inflammation [36]. It has both cutaneous and musculoskeletal symptoms. The former encompasses psoriasis and nail lesions, while the latter involves arthritis, spondylitis, and enthesitis (inflammation of tendons and ligaments) [37].

AS, similar to RA, is an autoimmune rheumatic disease with similar clinical and genetic characteristics. It initially attacks the tiny joints of the hands and feet and primarily affects the spinal joints, sacroiliac joints, and the soft tissues around them such as tendons and ligaments [38]. Severe cases can lead to deformity and fusion of the spine, making the spine unable to bend normally. The etiology and pathogenesis of AS are very complex. Current studies have shown that heredity is the key factor in the occurrence of AS, and the abnormality of human leukocyte antigen-B27 (HLA-B27) is an important reason individuals are prone to AS [39].

ReA is inflammatory arthritis that occurs after gastrointestinal or genitourinary tract infection. Infection, HLA-B27 anomalies, and immunological diseases are related to the development of ReA [40]. The common clinical manifestations of ReA are fever, bacterial infection, and acute asymmetric joint inflammation, mainly involving the knee, ankle, and other large joints of the lower limbs, which is characterized by joint swelling, pain and effusion [41]. Together with PSA and AS, they are included in the category of seronegative spondyloarthropathies [42].

## 3. Drug Delivery Strategies for Arthritis

Although different types of arthritis have different causes, they all show inflammation of joints. The common goal of treatment for arthritis is to alleviate pain, reduce inflammation, and slow joint degeneration to prevent disease progression and joint deformity or impairment. Therefore, the therapeutic drugs for different types of arthritis have certain similarities. In this section, we summarized the various types of drugs and their delivery methods, as well as the potential of MNs in their application. The drugs that have been approved for the treatment of arthritis as of 20 April 2022 are summarized in Table 2. As we can see in Table 1, there are various drugs for arthritis treatment, and each one has a different delivery route. Therefore, MNs may be a certain vehicle not only for loading various types of drugs, but also enhancing the therapeutic effects for its diversity.

### 3.1. Oral Drugs

In the first-line treatment of arthritis, oral NSAIDs are used to reduce pain and inflammation rapidly [23]. NSAIDs block the synthesis of prostaglandins through cyclooxygenase-1 and cyclooxygenase-2 to achieve analgesic and anti-inflammatory effects. For example, acetylsalicylate (aspirin) and naproxen (Naprosyn) can reduce pain and joint stiffness but do not interfere with the factors causing the joint injury. Thus, they can only alleviate symptoms but not cure diseases [5]. However, the cardiovascular and gastrointestinal side effects of NSAIDs have been confirmed, and long-term oral administration of NSAIDs is not recommended in treating arthritis [43].

Some conventional synthetic DMARDs are also used to treat various types of arthritis. Because of their effective anti-inflammatory and immunomodulatory ability and good tolerance, DMARDs have been regarded as the first choice for the treatment of arthritis. Methotrexate (MTX), a small-molecule immunosuppressive agent, is a folate antagonist responsible for inhibiting the synthesis of purine, pyrimidine and polyamines, and adhesion of inflammatory molecules [44]. However, long-term toxicity is unavoidable due to the anti-folate effect of MTX. The common side effects are gastrointestinal reaction, hepatorenal toxicity, and bone marrow deterioration [45]. Compared with oral dosage forms, drugs delivery via MNs can improve the efficiency of drug absorption and reduce the gastrointestinal reactions and systemic side effects associated with long-term oral administration.

### 3.2. Injections

For some drugs which are not suitable for oral administration such as glucocorticoids (GCs) and biological drugs, e.g., TNF-α inhibitors, injection may be a more appropriate way of administration. GCs are anti-inflammatory drugs that act faster than NSAIDs. They inhibit the development of inflammation by activating glucocorticoid receptors and inhibiting pro-inflammatory transcription factors [46]. Since GC receptors exist in practically every human tissue and have pleiotropic effects on several signal pathways, GCs can cause a wide range of negative consequences. Their side effects involve most organs throughout the body, including musculoskeletal, gastrointestinal, cardiovascular, and endocrine systems [47]. Therefore, to alleviate local inflammatory symptoms, it is recommended to use low-dose intra-articular GC only when the symptoms of arthritis are severe.

TNF-α is a cytokine that can promote joint inflammation. TNF-α inhibitors can prevent the release of TNF-α-driven inflammatory factors and the further development of inflammation [48]. The majority of TNF-α inhibitors are monoclonal antibodies or fusion proteins. Therefore, their efficacy needs to be guaranteed by injection administration. TNF-α inhibitors etanercept, adalimumab, and infliximab are commonly used to treat arthritis. These drugs have different ways of administration. Adalimumab and etanercept are typically administered subcutaneously, and infliximab can be administered intravenously or via intra-articular infusion [21]. Although TNF-α inhibitor treatment has shown improved therapeutic outcomes in various types of arthritis, the injection delivery of biological drugs will lead to increasing the risk of infection and decreasing patient dependency [49], which sets a barrier to the long-term delivery of TNF-α inhibitors. This is despite the fact that injectable administration shows high delivery efficiency while avoiding gastrointestinal reactions. However, the need for professional administration and patients’ fear of needles still greatly limit the clinical use of injection administration. Hence, as a transdermal delivery method, MNs are not only convenient to administer, but also can minimize the pain caused by treatment.

### 3.3. Transdermal Application

Due to the gastrointestinal and cardiovascular adverse effects of NSAIDs, topical NSAIDs have become a potential alternative to oral administration. The FDA approved two topical NSAID formulations for OA in 2007: diclofenac sodium 1% gel and diclofenac sodium 1.5% in 45.5% dimethylsulfoxide [50]. Topical NSAIDs showed a higher safety profile than oral forms due to less systemic absorption. Hence, the American College of Rheumatology (ACR) recommended topical NSAIDs rather than their oral forms for knee OA patients aged 75 years or older [51].

Chinese external therapy (CET) is a topical application method using Chinese herbal medicines as the main ingredients. It has been used in Chinese medicine for thousands of years to relieve arthritis symptoms. *Tripterygium wilfordii* is commonly used to alleviate the symptoms of arthritis in China, but oral administration has been limited in clinical application because of its hepatotoxicity and reproductive toxicity [52]. *Tripterygium wilfordii* Hook F gel, made from *Tripterygium wilfordii* extract, is for topical use in hospitals [53], which can relieve arthritis with reduced systemic toxicity. However, the biggest limitation of traditional transdermal drug delivery is the low drug delivery efficiency. MNs can easily break through the skin barrier, thus achieving efficient drug delivery, which makes MNs the best solution to the limitations of traditional transdermal drug delivery.

## 4. MN Drug Delivery System

Most NSAIDs and synthetic small-molecule DMARDs require long-term administration to treat arthritis, known to cause relevant side effects. Most corticosteroids and biologic drugs require constant injections, and long-term injections not only cause irreversible damage to skin vessels and other tissues but also increase the patient’s distress. Furthermore, due to the skin barrier or articular cavity barrier, some water-soluble or macromolecular drugs cannot be delivered efficiently by transdermal administration. MNs have the advantages of both non-invasive (topical transdermal) and invasive (injectable) drug delivery methods, which can overcome the main limitations mentioned above. MNs are considered topical transdermal, and drugs delivered by MNs are free from first-pass effects, which avoids gastrointestinal reactions. Moreover, MNs are easily handled by patients, providing efficient drug delivery. Thus, MNs have become a new generation of topical drug delivery system for arthritis drugs [54].

### 4.1. Types of MNs

MNs can be classified based on component materials, preparation methods, applications, or designs. In regard to their designs, the types of MNs widely studied today include solid MNs, hollow MNs, coated MNs, dissolving MNs, hydrogel-forming MNs, and other new types [54,55]. The drug delivery mechanisms of various types of MNs are described in Figure 1, and the summary of the advantages, disadvantages, and mechanisms of these MNs are listed in Table 3.

#### 4.1.1. Solid MNs

The presence of a stratum corneum lipophilic barrier makes it suitable to deliver only drugs with logP in the range of 1.0–3.0 and small molecules (molecular weight of >500 Da) via topical transdermal administration [78]. Solid MNs can penetrate the skin to form microchannels. After removing MNs, the drugs in the forms of lotions or patches can be administered, which will pass through the outer stratum corneum via the micron-sized pores and be absorbed by the skin to generate local or systemic effects [63]. Thus, solid MNs are usually not used alone but as an adjunct for transdermal therapeutic drugs. Solid MNs are made of materials such as silicon, titanium, nickel, or ceramics without voids inside [54]. Silicon MNs are mostly prepared by the silicon dry etching process [79]. Metal MNs, such as titanium and nickel, are mainly prepared by 3D laser ablation, laser cutting [80], and dry and wet etching [81] methods. Ceramic MNs are prepared by ceramic microforming and sintering methods. Solid MNs have a wide range of applications in the treatment of arthritis because they can be directly applied to existing transdermal preparations as an auxiliary device to improve the efficiency of drug delivery.

#### 4.1.2. Hollow MNs

Hollow MNs are made of metal or silicon, with a certain space inside and an opening tip of the MN [54]. When hollow MNs are inserted into the skin, the drug is infused or diffused through the opening tip of the MNs, similar to subcutaneous injection [56]. Since many arthritis therapies that require injections need to be applied over a long period of time, it becomes important to improve patient compliance. Because of the shorter needle of the hollow MNs, it is more acceptable to patients than subcutaneous injections. In the same way as solid MNs, hollow MNs can be used as an adjunct delivery device to existing arthritis treatment drugs. Compared to solid MNs, hollow MNs, coated MNs, and dissolving MNs allow delivery of larger doses of drugs. Moreover, hollow MNs, coated MNs, and dissolving MNs can be combined with other techniques, such as pressure and ion introduction, for continuous drug delivery [82,83]. Microelectromechanical systems (MEMS) technology is often used for the preparation of hollow MNs [56].

#### 4.1.3. Coated MNs

Coated MNs are derived from solid MNs, in which a drug or a vaccine is coated on the surface of the solid MN [59]. When coated MNs are inserted into the skin, the coating drug dissolves and enters the skin [56]. The faster the coating dissolves, the faster the drug releases. Therefore, water-soluble formulations are often used for coating drugs, which are more effective for drug diffusion. Surfactants can also be added to the formulations to promote faster wetting of the MN surface [57]. Thus, coated MNs are suitable for steroids to provide anti-inflammatory and analgesic relief or for pain relievers, which will generate pharmacological effects quickly in small doses. The drug is usually attached to the surface of the MNs by dipping, spraying, or coating, and the thickness of the coating determines the amount of drug loaded onto the MNs [12].

#### 4.1.4. Dissolving MNs

The needles of dissolving MNs are made of water-soluble or biodegradable biomaterials that encapsulate drugs [84,85]. After inserting into the skin, the needles of dissolving MNs are completely dissolved, and the drug is released into the skin [63]. Dissolving MNs are an environmentally friendly solution, which can be easily handled by patients and leave no dangerous or wasteful products behind, such as metal needles or glass. Dissolving MNs can improve the quality of life of patients with arthritis that need frequent injections. A wide range of polymer materials with different properties can be applied to develop dissolving MNs based on the need. Thus, dissolving MNs have the potential to deliver various drugs with complex properties. The molecular weights of the polymer materials used for the preparation of dissolving MNs should be lower than 60,000 Da to avoid excessive accumulation of polymer materials in the body, which may cause invisible damage [85,86]. Dissolving MNs are mainly prepared by micromolding technology. The conventional micromolding operation is to add the mixture of drug and matrix (carrier) directly into the mold and wait for it to dry. During the preparation, after filling the mold with the mixed solution of drug and matrix material, the outside of the mold needs to be scraped off to ensure a homogeneous shape of the MNs, but this operation causes drug waste. Therefore, many dissolving MNs are prepared in two parts: first filling the tip portion of the MN’s negative mold using a matrix solution containing the drug, and then filling the remaining portion of the mold as the backing layer using a matrix material which does not contain the drug [14].

#### 4.1.5. Hydrogel-Forming MNs

Hydrogel-forming MNs, first proposed in 2012, are formed by swellable hydrogel polymers [68]. After inserting into the skin, hydrogel-forming MNs rapidly absorb the interstitial fluid and swell but do not dissolve, forming a hydrogel state with multiple microchannels. Then, the drug can enter the skin through the channels [67]. Since the hydrogel-forming MNs are usually made of biocompatible polymers and have insoluble properties, they have good biocompatibility while leaving no matrix material in the body, avoiding the risk of long-term accumulation of matrix material. Furthermore, hydrogel-forming MNs have the capacity for loading high doses of drug, and the drug release rate can be regulated by the degree of cross-linking of hydrogel. Thus, hydrogel-forming MNs become an important solution for the treatment of arthritis requiring continuous anti-inflammatory analgesia [87]. The preparation of hydrogel-forming MNs requires physical or chemical cross-linking of polymers to form hydrogel states. Then, MNs are shaped by micromolding or spin-casting [67,87].

#### 4.1.6. Other Novel MNs

Stimulus-responsive MNs are a new type of MNs that can release loaded drugs in a targeted or timed manner according to pH, pathological conditions, or external stimuli (light, magnetic, etc.) [88]. At present, the research on stimuli-responsive MNs mainly focuses on pH-responsive, enzyme-responsive, glucose-responsive, and light-responsive MNs. The materials of MNs are selected according to the specific response to the stimulus [89].

The bionic MNs developed so far mainly include mosquito-like MNs [90], North American porcupine feather-like MNs [43], caterpillar-like MNs [44], and bee-sting MNs [91]. The imitation bee-sting MNs are prepared by imitating a bee sting and can be used for tissue adhesion, transdermal drug delivery, or biosignal recording. The preparation of bee-sting MNs is divided into two steps. The first step is to prepare the parent MNs, and the second step is to form inclined MNs on the surface of the parent MNs [55].

### 4.2. Requirements and Design of Geometry and Mechanical Strength of MNs

The essential factors for designing MNs include the shape, length, tip diameter, and needle spacing (as shown in Figure 2). These parameters determine the efficacy, integrity, drug dosage, and degree of pain of the MNs [92]. For example, needles of inappropriate length or shape can cause erythema, edema, or damage to the skin, resulting in pain [93]. The mechanical strength of MNs is very crucial. MNs that do not meet the mechanical strength requirements will easily break during administration, which may leave part of the MNs in the skin and cause invisible damage. Consequently, this may affect the delivery of the correct drug dose [18]. The design requirements are varied according to the type of MN, and the specific requirements for different MNs are described in the following.

#### 4.2.1. Geometry of MNs

The geometry of MNs has a significant impact on the performance of MNs, especially in terms of mechanical strength and the sensation produced during application. Shapes of MNs include cones, cylinders, pyramids, crosses, etc. Studies have confirmed that the cone shape has a strong penetrating ability and especially has better curative effects on immune diseases, and can be easily prepared and removed from the skin [58,94,95]. In addition, the lumen of the hollow MNs is preferably made into an “I” shape, whose strength is stronger than that of the traditional round lumen [96].

The length of the MN is generally 25–1500 μm. Theoretically, an MN with a length of 25 μm is enough to penetrate the 10–20 μm thick stratum corneum to form a pathway and deliver the drug into the cortex. When the MN length exceeds 120 μm, it can pass through the dermis, which is full of nerves and capillaries, and the drug can enter the systemic circulation through the capillaries [97,98,99]. However, in practice, shorter MNs (less than 300 μm) have difficulty penetrating the SC smoothly into the viable epidermis due to the elasticity of the skin [100]. Therefore, most of the existing studies on drug delivery MNs are 300–800 μm in length [98]. Because the thickness of the joint capsule is about 580–730 μm [101,102], for intra-articular drug delivery to penetrate the joint capsule, the length of MNs is usually 800 μm. It is important to note that longer MNs can not only enter deeper into the epidermis and irritate nerve endings, causing pain, but also have the risk of breaking during application. For this reason, MNs longer than 800 μm are less frequently studied. In addition, delivering certain drug doses may require a large number of MNs per unit, which can cause excessive pain. Therefore, the length and quantity of needles per unit area are the main factors to be weighed in the design of MNs [87,103]. The diameter of the tip of MNs is generally 1–25 μm [104,105]. To ensure a good puncture effect, it is better designed to be less than 15 μm [106,107,108,109]. A recent study has shown that the mechanical strength increases as the aspect ratio of the MN decreases, with the most suitable aspect ratio for clinical use being 2:1 [110]. Others suggest that cross sections of MNs should be as small as possible to maintain a low loading force for an easy insertion [98,111].

Moreover, there are two crucial aspects for the development of MNs. The first one is the edge-to-edge distance of the hydrogel-forming MNs which, generally, should be greater than 50 μm [87]. If the needles are too close together, the MNs will collide with each other during the expansion process, thus disrupting the MN arrays and affecting drug delivery [112]. The other is the tip of hollow MNs, which should retain an opening on its side to allow fluid to flow smoothly and continuously [113,114].

#### 4.2.2. Mechanical Strength of MNs

The mechanical strength of MNs can be tested with tensiometers, strain gauges, or texture analyzers by placing the MN patch on the instrument platform for compression. The maximum force that the tip of the MNs can withstand is the parameter of its mechanical strength [115,116,117,118]. The force that the MN tip can withstand should be greater than 0.66 N. Otherwise, it will not be able to pierce the skin [110,119,120,121]. For dissolving MNs with low mechanical strength, high-viscosity materials such as PVA can be used as the matrix. Water-soluble substances such as polysaccharides can be added to improve the mechanical strength of dissolving MNs [77,122,123,124,125]. It is also possible to add a cross-linker (citric acid, I 2959) to a cross-linkable material (PVP, PVA) to allow the material to achieve higher mechanical strength through chemical or physical cross-linking [115,126].

## 5. Recent Advancements of MNs in Arthritis Treatment

In recent years, MNs have emerged as one of the most promising delivery strategies for arthritis drugs in laboratory studies. Due to their variety and characteristics, various types of MNs and delivery enhancement modalities could be explicitly applied to deliver different drugs. This section lists laboratory studies that have emerged in recent years on the administration of arthritis drugs by MNs (Table 4).

### 5.1. Solid MNs

As drug-free medical devices, solid MNs have been developed rapidly due to their simplicity of preparation. Solid MNs only serve to open the skin orifice, and the drug in a transdermal formulation is delivered after the MN is removed. The lipophilic barrier of the skin makes it difficult for hydrophilic MTX (logP-1.85) to passively penetrate into the body. Mehtab Jabla et al. used solid MNs to form hydrophilic microchannels across the skin, which provided the passive permeation of MTX [128]. The lag time of the average cumulative amount of MTX was shortened from 8 h to 2 h, indicating a faster permeation rate of hydrophilic drugs after MNs were used [128]. In addition to hydrophilic small-molecule drugs, macromolecules are also difficult to permeate through the skin barrier. Melittin has been investigated extensively in the treatment of RA inflammation and pain. Since it is an active macromolecule (2840 Da), even with the transdermal formulations, melittin is unable to transfer into the body effectively by topical administration. Mengdi Zhao et al. prepared a gel formulation of melittin and used solid MNs to deliver it into the skin [130]. This study showed that changing MN parameters can form various degrees of microchannels in the skin, which affects the efficiency of drug delivery [130]. A shorter length of MNs, lower application force, and shorter application duration were shown to reduce drug penetration [130].

Although the application of solid MNs is relatively limited, they can be used to topically deliver different drugs by optimizing the formulations. For example, encapsulated liposomes, ethosomes, and nanoparticles can further promote drug penetration; encapsulated nanoparticles or microsponge gels can induce long-lasting drug release combined with solid MNs [129,131,132,154,155].

### 5.2. Hollow MNs

Conventional hollow MNs deliver drugs in a manner similar to subcutaneous injection. After the insertion of hollow MNs into the skin, the drug enters the body through holes in the interior of each needle. Hollow MNs are a better choice for delivering high-molecular-weight drugs such as proteins and monoclonal antibodies (biological DMARDs) [93]. Bushra et al. investigated the ameliorative effects of denosumab on osteoporosis by delivering the drug via 3D printed hollow MN arrays. Denosumab is the FDA-approved treatment for osteoporosis, and can be used to treat RA [156]. The denosumab concentrations in the plasma of mice of the SC group were compared to the hollow MN group over 28 days to determine the release profiles (blood concentration/time) of each route of administration. The result showed that both groups had similar drug release profiles in 28 days, indicating that the hollow MN arrays not only had the ability to deliver drugs as effectively as the subcutaneous injection, but also had the advantages of being minimally invasive and pain-free stimuli [134].

Recently, a novel hollow MN for the delivery of biological drugs has been reported by Carcamo et al. [135]. These hollow MNs contained an external hydrogel-forming MN shell and an internal hollow cavity, which allowed for direct loading of drug powder in the cavity. Thus, it increased the drug loading capacity of the MNs, and the loaded drug is much more stable in the solid form. At the same time, adding water-soluble modifiers such as NaCl to the hydrogel matrix of the shell can form a porous structure after removing MNs, which will accelerate the drug release. Carcamo et al. investigated the synergistic effects of hollow MNs on the intradermal administration of the JAK inhibitor tofacitinib compared to the cream form of tofacitinib. The maximum release rates of the control cream and NaCl hollow and dissolving MN arrays were 27.86% and 44.31%, respectively. Compared to the control ointment, hollow MNs slightly increased the drug release rate and improved the stability of tofacitinib. This kind of MN also provides a new direction for the delivery of biological drugs, but its drug release rate is lower than that of dissolving MNs [135]. In the future, the design of this kind of MN should focus on the modifiers that can improve the drug release rate.

### 5.3. Coated MNs

Coated MNs have the characteristics of rapid drug release and precise dose control. Since coated MNs can only deliver the drug being loaded on the surface of the MNs, they have limited drug loading capacity, which is applied to deliver drugs with strong pharmacological efficacy in a low dosage, such as vaccines or analgesics [60,157]. Most arthritis drugs such as NSAIDs and conventional synthetic or biological DMARDs have a slow onset of action and require a larger dose, which is not suitable for coated MNs. Therefore, compared to other types of MNs, there are fewer studies on the application of coated MNs for the treatment of arthritis. Abdalla et al. investigated the anti-inflammatory and analgesic effects of opioid tramadol delivered by coated MNs. They used the dipping method to precisely coat the MNs with different concentrations of tramadol hydrochloride. An amount of 1% (*w*/*v*) carboxymethyl cellulose was added to the coating solution to enhance the viscosity and 0.5% (*w*/*v*) Lutrol F-68 NF was added as a surfactant, resulting in a better coating of the drug. Compared to joint injections, the duration of analgesia delivered by coated MNs increased from 2 h to 2 days. In addition, intra-articular injection of tramadol significantly reduced the levels of the pro-inflammatory cytokines TNF-a and IL-1b and increased the expression of anti-inflammatory cytokine IL-10. However, the levels of TNF-a, IL-1b, and IL-10 were not significantly different from the non-treatment group after 2 h, indicating that the duration of the anti-inflammatory effects produced by intra-articular tramadol injection was short. Contrarily, the treatment of tramadol delivered by coated MNs reduced the levels of TNF-a and IL-1b and increased the level of IL-10 up to 6 days [158]. This study demonstrates the great potential of coated MNs as an option for the delivery of pain medication.

### 5.4. Dissolving MNs

Dissolving MNs are known for their high drug delivery efficiency compared to various other MNs due to the large amount of biocompatible matrices that can be used to develop dissolving MNs. Therefore, dissolving MNs currently account for the majority of MN research in delivering arthritis therapeutics.

The basic dissolving MNs are prepared by mixing a single polymer material with a drug solution; the preparation process is simple and convenient. Korkmaz et al. used carboxymethylcellulose (CMC) to prepare solubilized MNs encapsulated with TNF-α inhibitors by using the micromilling/spin-casting fabrication method at room temperature. Due to the continuous and simple preparation process, 500+ MN arrays could be prepared within 6 h, which is expected to be easily scaled up for automated industrial production [137]. Although the MNs built with single polymer material are more homogeneous in nature, it limits the application for various drugs. The performance of basic dissolving MNs can be improved by modifying the single matrix with functional groups or by using several matrices together with different proportions. Hyaluronic acid (HA) is one of the most suitable polymers for the preparation of dissolving MNs due to its high biocompatibility, bioequivalence, and hydrophilicity. Nevertheless, due to the limit of its mechanical strength, MNs developed by HA may not be strong enough to insert too deep into the skin [159]. Gu’s group used methacrylic anhydride to modify HA and obtain an acrylate-modified HA, which can be cross-linked by UV to improve the strength of the dissolving MNs [160]. The dissolving MNs prepared by this method were successfully loaded with etanercept (TNF-α inhibitor) and had sufficient mechanical strength to penetrate the skin, and could deliver the drug to the capillaries [142]. Similarly, Du’s group used methacrylate to modify HA and obtain MeHA. The MeHA MNs were cross-linked by UV light and loaded with melittin, which can be used for the controlled release of the drug. The in vitro release experiment showed that the unmodified HA MNs released all loaded melittin within 10 min, while the MeHA MNs could induce the in vitro release for 480 min and the in vivo release for up to 7 days [67].

Different polymers have different properties. Thus, the properties of MNs can be optimized by using different polymers. Amodwala et al. used hydrophilic polyvinyl alcohol (PVA) and polyvinylpyrrolidone (PVP) as the matrix to make fast-dissolving MNs loaded with meloxicam [141]. PVA provided strength to the MNs and also acted as a solubilizer to facilitate the release of the insoluble meloxicam loaded in the MNs. PVP acted as a highly hydrophilic filler for the MNs, dissolving rapidly to create hydrophilic channels for rapid delivery of the drug. This dissolving MNs had the maximum axial needle fracture force of 0.9 N when the ratio of PVA to PVP was 9:1 with 50% solid content. The PVA–PVP MNs could rapidly release all meloxicam in 60 min. Some types of arthritis, such as PsA, have both psoriasis and arthritic symptoms, which will require the administration of several medicines at the same time. Therefore, another new type of MNs with the ability to deliver multiple drugs has emerged. Yu’s group designed a layered dissolving MN that can load two drugs of different properties and deliver them to different depths under the skin. They divided the MNs into a tip layer, an inter-layer, and a pedestal. The tip layer was designed to deliver the hydrophilic NSAID diclofenac (DIC) into the joint cavity to relieve joint inflammation. The inter-layer was designed to treat psoriasis with the hydrophobic immunosuppressant tacrolimus (TAC). The tip and inner layers were optimized with different ratios of PVA and PVP. Since they were all hydrophilic substrates, the DIC could be easily loaded into the tip layer. At the same time, TAC was loaded into the inter-layer by the addition of nicotinamide to solubilize TCA and improve the mechanical strength of the inter-layer. In vitro and in vivo skin penetration experiments showed that the tip layer DIC was delivered into the joint cavity, while the inter-layer TAC was retained on the skin surface with significantly higher delivery of both drugs than the one-layer MNs loaded with a mixture of TAC and DIC. This layered dissolving MN provided an effective and feasible strategy for arthritis comorbidities that needed to be treated with multiple drugs [148].

Nanocarriers can be added to dissolving MNs to enhance drug delivery. Shende et al. prepared poly (lactic-co-glycolic acid) (PLGA) microspheres encapsulated with folic acid using a multiple-emulsion system (w/o/w type). The microspheres were loaded into methotrexate-containing dissolving MNs. Thus, the MNs can co-administrate folic acid and methotrexate into the skin and achieve synergistic effects. Moreover, PLGA microspheres have a controlled release effect. In vitro release studies showed that more than 80% of methotrexate and nearly 60% of folic acid are released from the MN array within 8 h, followed by a slow controlled release over 24 h, which results in longer-lasting effects of both drugs [64]. Hu et al. applied a more functional PLGA microsphere to develop solubilized MNs. Tetrandrine, an anti-arthritic medication with limited solubility, was loaded into PEGylated multi-arm PLGA microspheres, which enhanced the solubility of poorly soluble drugs and also had immune escape functions and acid-responsive features. The acid-responsive release of the drug was simultaneously achieved by using calcium carbonate hybrid nanocarriers. The multifunctional microspheres were loaded into the dissolving MNs, which were prepared from peach gum with higher mechanical strength and better physical stability compared to the MNs prepared from hyaluronic acid [149]. To modify the solubility of lipid-soluble drugs in hydrophilic dissolving MNs, the inclusion complex was also commonly applied. Chen et al. used hydroxypropyl-β-cyclodextrin inclusion complexes to encapsulate the lipophilic polydatin to treat GA. The solubility of polydatin in water was enhanced from 0.357 mg/mL to 124.47 mg/mL after being encapsulated by the inclusion complexes. Since the inclusion compound made the lipophilic drug much more water-soluble, it could be effectively loaded into water-soluble polymer-based dissolving MNs [145]. Liposomes are also widely used as drug nanocarriers, and have good biocompatibilities to reduce the side effects of drugs. Zhou et al. developed a liposome-loaded dissolving MN system which could efficiently load the poorly soluble triptolide on dissolving MNs and release it into the skin. This new MN improved the bioavailability of triptolide and reduced its systemic side effects [94].

In addition to the above innovations in MNs, the technologies for externally enhancing drug delivery, such as iontophoresis (ITP), can also be combined with MNs to improve drug penetration into the skin. ITP is an active energy-release process that uses microcurrents to promote the entry of ionic drugs into the body [161]. Vemulapalli et al. utilized ITP to drive the delivery of methotrexate, which was negatively charged under physiological PH conditions. Applying electrical repulsion in combination with soluble MNs showed a 25-fold increase in the in vivo delivery of methotrexate compared to MNs or ITP alone [136].

### 5.5. Hydrogel-Forming MNs

Hydrogel-forming MNs have been widely used in recent years due to their long-lasting drug-releasing ability, high biocompatibility, and because they do not leave any residue in the skin [162]. In a study by Cao et al., a hydrogel-forming MN was used to deliver a modified DEK-targeting aptamer for the treatment of RA. In their initial study, dissolving MNs were also used to deliver the aptamer. However, due to the limited drug loading capacity and the hydrophilic layer of dissolving MNs, there was an incomplete release of the aptamer. Moreover, there were long-term safety issues with matrix residues left in the skin. They prepared a hydrogel-forming MNs with hydrophobic EC as the base layer, which showed a higher release rate of the aptamer. Since the hydrogel-forming MNs can be removed from the skin intactly after drug release, it reduces the risk of matrix residue-driven side effects for the long-term usage of MNs [115]. In a study by Chen et al., the hydrogel-forming MNs were loaded with MTX for treating RA. They combined a hydrogel-forming MN array with an MTX-loaded patch-like polymeric reservoir to increase the amount of drug being loaded on MNs. The dissolving MNs in the study of Vemulapalli et al. could deliver MTX at a rate of 18.2 μg·cm^2^/h. The HA dissolving MNs prepared by Du et al. showed a delivery rate of 13.8 μg/10 × 10 in their MN array for MTX. Compared to the above methods, the hydrogel-forming MNs combined with a drug reservoir developed by Chen et al. could deliver MTX at a high steady-state flux (506.8 ± 136.9 µg·h/cm^2^) and sustained a delivery rate of 6.8 ± 0.4 mg/cm^2^ of MTX for 24 h [126].

## 6. Translation of MNs from Laboratory to Clinic and Market

Because transdermal drug delivery has many irreplaceable advantages over other traditional drug delivery methods, the application of transdermal drug delivery in clinical settings is gaining overwhelming attention. It has great potential for market growth [14]. The global market for transdermal drug delivery is forecast to increase by nearly $1.79 billion from 2019 to 2023. MN systems, a new form of transdermal drug delivery, have moved from the laboratory to the clinic in recent years. A search of ClinicalTrials.gov using MNs as a keyword revealed 130 clinical trials worldwide, 85 of which have been completed, including two phase IV clinical trials. Most of these clinical trials are focused on anesthesia, vaccine delivery, skin diseases, and aesthetics [163,164]. Solid MNs, hollow MNs, and dissolving MNs are currently developed for clinical usage [165], which shows great potential for dosage form development.

There are still many challenges in developing approved MNs for clinical application from laboratory studies. First, the main obstacle limiting clinical translation is the relatively low drug loading capacity of MNs. The in vitro and in vivo models used in the laboratory require lower doses of drugs to show the therapeutic effects compared to the patients. Currently, some research is gradually beginning to address this issue by expanding the area of MN arrays or utilizing new manufacturing strategies to increase the maximum loading capacity of individual arrays, allowing MNs to carry sufficient doses of drug and deliver them to the human body [166,167]. In addition, the penetration ability of MNs into human skin should be evaluated more accurately. Because ex vivo human skin samples are more expensive, in vitro experiments mostly use newborn pig skin with similar mechanical properties to human skin. There are structural and biological differences between animal and human skin, which can lead to some deviations in accessing the MN penetration ability [168,169]. Therefore, the use of optical coherence tomography (OCT) to image and measure the real skin penetration ability of MNs in vivo prior to clinical application can avoid the risk of breaking MNs when applied to human skin [97]. Besides the penetration ability, the degree of pain caused by MNs could not be accurately determined in the laboratories. Generally, MNs cause less pain than subcutaneous injections, but different results may occur in different patients [92]. A clinical study of the intradermal administration of adalimumab by hollow MNs in 2021 showed that hollow MNs produced higher pain compared to conventional subcutaneous injections, but the drug delivered by hollow MNs had a relatively higher bioavailability. The investigators suggested that this may be related to the dose delivered. Another study showed that the pain caused by MNs was not only related to the geometry of MNs, but also to the nature and amount of fluid injected into the skin [170]. Thus, a comprehensive estimation of the penetration ability of MNs and the pain caused by MNs in humans are needed for the development of MNs.

In general, MN array patches for drug delivery are currently under development and are still in the early stages of manufacturing for commercial applications. The biggest hurdle in scaling up MN production from the laboratory to the factory is establishing a standard manufacturing process [171]. The first pharmaceutical MN array patch product to seek regulatory approval in the United States is the MN array (Qtrypta™) containing the migraine drug zolmitriptan, developed by Zosano [172]. However, different MN patches produced different drug exposure levels when used on study subjects, which became a hindrance in their new drug application process. In addition, the sterility required for MN array products has not been studied or standardized. Many arthritis drugs are biologically active and unstable under sterilization. The development of suitable sterilization methods with low cost for MNs should be considered for scaling up the manufacturing of MNs [171]. There is still a great deal to explore to transform MNs from research results into products.

Up to now, the application of MNs in delivering arthritis drugs has mostly been in the laboratory research stage, and only one MN have been tested in clinical trials (Table 5). However, some clinical studies which are widely developed, delivering analgesics, anti-inflammatory drugs, and bioproteins using MNs, present great reference value for MN delivery of arthritis therapeutics from laboratory to clinical applications [124]. Due to the specific location of arthritis, the mobility of the joint should be considered while delivering some drugs that require local intra-articular application using MNs. When applying MNs at the joint, the movement of the joint may cause parts of the MN patch to buckle, fall off, or even break. Solid MNs, coated MNs, and hollow MNs can avoid this issue due to their rapid drug delivery process. Dissolving MNs and hydrogel-forming MNs are designed for the prolonged release of drugs, and it is recommended to add auxiliary parts such as straps to help stabilize these MNs in the clinic.

## 7. Conclusions and Future Perspectives

Arthritis occurs in a wide range of people of different genders and ages. It seriously affects the quality of life of a lot of patients. The current drugs for arthritis are mostly administrated by oral or injection routes, which will need long-term treatment. The gastrointestinal side effects and patients’ fear for needles have prompted the development of other arthritis treatment strategies, including transdermal drug delivery technology. As an emerging technology in the field of transdermal drug delivery, MNs have quickly attracted the interest of researchers due to their variety and potential functions. MNs have shown significant advantages in the delivery of biological drugs or drugs that require local or frequent injections, filling the gap in traditional delivery methods for arthritis drugs. Specifically, the use of biocompatible polymers allows for the stable loading of biologic DMARDs and effective delivery. Hydrogel-forming MNs can be used to obtain a slow release of drugs, which reduces the frequency of drug injections. Fast-dissolving MNs can be used for safely and rapidly loading precise doses of steroids into the joint to relieve inflammation, which also greatly reduces the systemic responses of steroids.

To date, most of the studies on MN administration have focused on the treatment of RA, but since other arthritis have similar symptoms and drugs, the experimental results and development experience can also provide an important reference for the development of drug-loaded MNs in the therapy of other types of arthritis. In addition, arthritis is an inflammatory disease with typical characteristics; thus, MN studies in arthritis also provide valuable ideas for the therapy of other types of inflammatory diseases. Furthermore, most studies of MNs on delivering arthritis drugs are undergoing research in laboratories, and have not been applied in the clinic. Therefore, more clinical studies to determine the effectiveness and safety of MNs are needed. There is a lack of study on the geometry and mechanical strength of MNs that are suitable for topical use in joints. To deliver drugs into the joint cavity, MNs will need to penetrate the skin and the joint cavity, which requires a certain length and mechanical strength of MNs. However, long MNs may cause pain and increase the risk of MN breakage in patients. Therefore, it is important to balance the penetration ability with safety and reduce the pain in designing MNs for administrating the drug to local joints. In addition, the movement of the joint may cause needle breakage or dislodgement of the locally applied MNs, resulting in intradermal injury. Thus, we believe that rapid-dissolving MNs are the most suitable type of MNs for topical administration at the joint due to their rapid dissolution feature, which shortens the administration time. Moreover, even if the MN needles break within the skin, the residuals of dissolving MN needles can be dissolved on their own without causing further damage to the skin. For patients, dissolving MNs can be applied by patients without needing to visit the clinic, which has great commercial potential. For drugs, small therapeutic doses of water-soluble drugs are most suitable for the microneedle’s transdermal delivery route due to the limitation of the lower drug loading capacity of MNs and the fact that the matrix of MNs is mostly hydrophilic. Although the FDA has established some regulations for MN products, the manufacturing guidelines and standards of MNs need to be further improved to enable this new technology to be applied to patients as soon as possible.

## Figures and Tables

**Figure 1 pharmaceutics-14-01736-f001:**
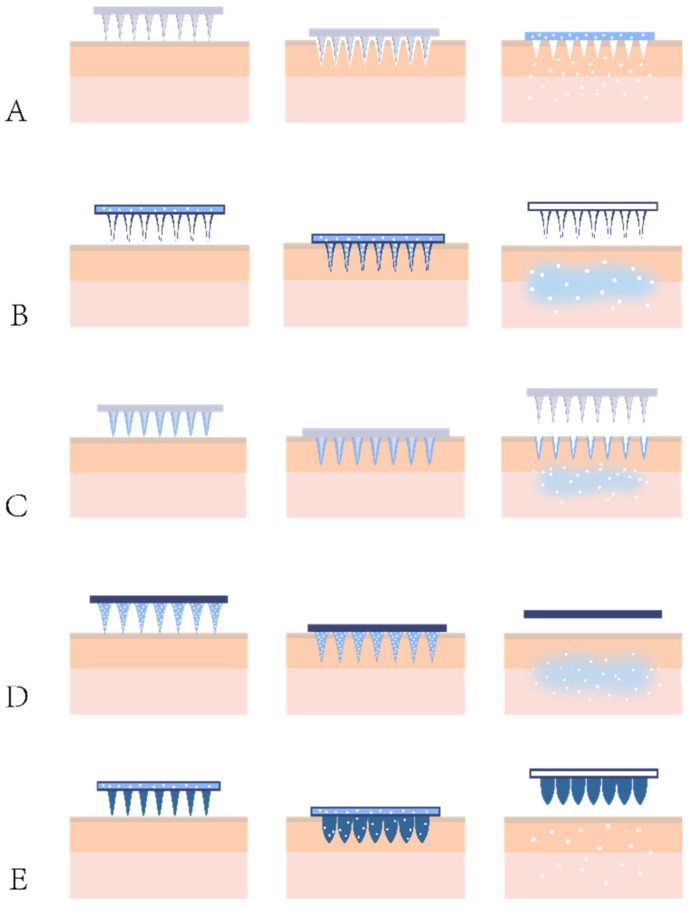
Drug delivery mechanisms for various types of MNs. (**A**) Solid MNs: Used for skin pretreatment, forms microchannels and is then removed, followed by the use of a topical formulation containing the drug for delivery of the drug through the microchannels, (**B**) Hollow MNs: After insertion into the skin, the drug loaded in the internal cavity of the MNs flows out, (**C**) Coated MNs: After insertion into the skin, the drug film wrapped in the outer layer of the MNs is released into the body, (**D**) Dissolving MNs: After insertion into the skin, the MN matrix dissolves into the body along with the drug encapsulated in it, (**E**) Hydrogel-forming MNs: After insertion into the skin, the MNs expand to form microchannels inside but does not dissolve, and the drug enters the body through the microchannels.

**Figure 2 pharmaceutics-14-01736-f002:**
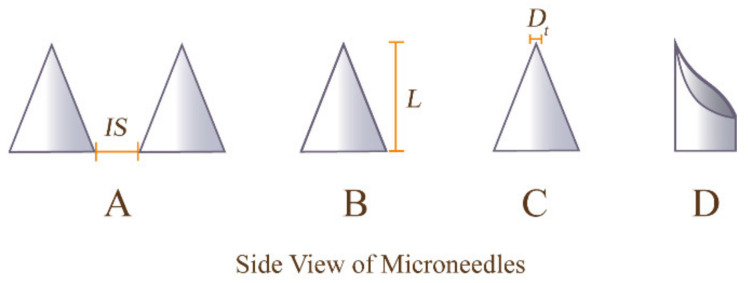
Schematic of MN geometry. (**A**) Edge-to-edge distance of MNs, (**B**) Length of needles, (**C**) Diameter of the tip, (**D**) Shape of MNs.

**Table 1 pharmaceutics-14-01736-t001:** Features of some common arthritis.

Type of Arthritis	Patient Age Group	Cause
RA	All ages	Synovitis caused by immune system diseases
OA	Over 60 years	Degenerative lesions with articular cartilage damage
GA	Over 40 years	Hyperuricemia and the deposition of MSU crystals in the joint capsule, bursa, cartilage, bone, and other tissues
JIA	Under 16	Unknown
PSA	All ages	Psoriasis
AS	10 to 40 years	Heredity (the abnormality of HLA-B27)
ReA	20 to 40 years	Gastrointestinal or genitourinary tract infection

**Table 2 pharmaceutics-14-01736-t002:** Summary of drugs for arthritis treatment.

Drug Classification	Active Pharmaceutical Ingredient	Type	Indications	Route
NSAIDs	Acetylsalicylic acid, Celecoxib, Choline magnesium trisalicylate, Diflunisal, Etodolac, Etoricoxib, Fenbufen, Nabumetone, Oxaprozin, Sulindac, Tiaprofenic acid, Tolmetin, Valdecoxib	Small-molecule	Arthritis	PO
	Benzydamine, Bufexamac, Etofenamate, Flufenamic acid, Salicylic acid			Topical
	Dexketoprofen, Flurbiprofen, Ibuprofen, Ketoprofen, Meclofenamic acid, Niflumic acid			PO/Topical
	Aceclofenac, Diclofenac, Indomethacin, Naproxen, Piroxicam			PO/IM/Topical
	Lornoxicam, Tenoxicam			PO/IM/IV
	Meloxicam			PO/IM/IV/Topical
Corticosteroid	Prednisone, Cortisone acetate			PO
	Betamethasone, Hydrocortisone			Topical
	Prednisolone			PO/Topical
	Hydrocortisone succinate			IM/IV
	Methylprednisolone			PO/IV/SC/Intra-articular
	Dexamethasone			PO/IV/IM/Intra-articular/Topical
	Triamcinolone			PO/Intra-articular/IM/Topical
Analgesic drug	Fenoprofen			PO
	Capsaicin			Topical
	Acetaminophen			PO/IV
	Thiocolchicoside			PO/IM/Topical
Conventional synthetic DMARDs	Auranofin, Chloroquine, Hydroxychloroquine, Mycophenolate mofetil, Penicillamine		RA	PO
	Azathioprine, Cyclosporine		RA/PSA	PO/IV
	Methotrexate		RA/JIA/ PSA	PO/IV/IM/SC/Intra-articular
	Sulfasalazine		RA/JIA/ PSA	PO
	Leflunomide		RA/GA/ JIA/PSA	PO
	Sodium aurothiomalate		RA/OA/ JIA/PSA	IM
Calcineurin inhibitor	Tacrolimus		RA	PO/Topical
JAK inhibitor	Tofacitinib		RA/PSA	PO
	Baricitinib		RA	PO
	Upadacitinib		RA/PSA/AS	PO
Supplements	Chondroitin sulfate		OA	PO
	Glucosamine		OA	PO/IM
	Hyaluronic acid		OA	Intra-articular
	Curcumin		Arthritis	PO/Topical
TNF inhibitor	Adalimumab	Protein	RA/JIA/ PSA/AS	SC
	Certolizumab pegol		RA/JIA/ PSA/AS	SC
	Etanercept		RA/OA/ JIA/PSA/AS	SC
	Golimumab		RA/PSA/AS	SC/IV
	Infliximab		RA/OA/ PSA/JIA/AS	IV/Intra-articular
T-cell inhibitor	Abatacept		RA/PSA/ JIA	SC/IV
B-cell inhibitor	Rituximab		RA	SC/IV
IL 1 inhibitor	Sarilumab		RA	SC
	Anakinra		RA/OA/ GA /JIA	SC/Intra-articular
	Canakinumab		RA/GA/ JIA	Intra-articular
IL 6R inhibitor	Tocilizumab		RA/JIA	SC/IV
IL 12\23 inhibitor	Ustekinumab		PSA	SC
IL-17A inhibitor	Secukinumab		PSA	SC
IL 23 inhibitor	Risankizumab		PSA	SC
	Ixekizumab		PSA	SC
RANKL inhibitor	Denosumab		OA	SC

SC: Subcutaneous injection; IM: Intramuscular injection; IV: Intravenous injection.

**Table 3 pharmaceutics-14-01736-t003:** Advantages, disadvantages, and research stages of different types of MNs.

Types of MNs	Advantages	Disadvantages	Research Stages	Ref.
Solid MNs	Simple to manufacture. High mechanical strength.	It may break when inserted into the skin, resulting in part of the MNs remaining on the skin after removing the MNs, causing invisible damage. Two-step dosing, slightly cumbersome steps, and prone to germ infection before dosing and after insertion.	It is mainly used for pretreatment of drug administration. Leiden University Medical Center developed a solid MN skin patch vaccine for the treatment of COVID-19 in April and is now in interventional clinical trials (data from ClinicalTrails.gov website).	[12,18,56]
Hollow MNs	Controlled dose of drug delivery. Adjustable drug delivery rate. No restriction on the type of drug administered.	It may break when inserted into the skin, resulting in part of the MNs remaining on the skin after removing the MNs, causing invisible damage. The skin hole caused by the insertion increases the risk of skin infection. High manufacturing requirements and preparation cost.	Accelovance Inc developed hollow MNs for intradermal delivery of normal saline in 2013, which has completed clinical trials and has not yet been listed (data from ClinicalTrails.gov website, Yaozhi data).	[12,57,58]
Coated MNs	Simple manufacturing. Rapid drug release.	The maximum drug dose that can be loaded is only 1 mg, so it is only suitable for the administration of drugs with high efficacy or small required doses. The frictional part will remain on the skin surface, resulting in a difference between the actual dose and the theoretical dose. The coating itself will affect the sharpness of the needle, and there is a barrier to penetration. Used needles need to be discarded, causing waste and producing sharp waste that is not easy to dispose of. The skin hole caused by the insertion increases the risk of skin infection.	Coated MNs currently under experimental research include: insulin-coated MNs for the treatment of hyperglycemia, desmopressin-coated MNs for the treatment of enuresis in children, and MNs for the treatment of hepatitis C. DNA vaccine-coated MNs et al.	[18,58,59,60,61,62]
Dissolving MNs	Dissolution rate can be adjusted by changing the material and shape of the needle body. One-step drug delivery, simple process. A wide selection of needle materials with good biocompatibility. Needle parts are completely dissolving, leaving no sharp waste after use.	Uncontrollable drug release. Lower mechanical strength than other types of MNs.	Methotrexate combined with PLGA-dissolving MNs for the treatment of arthritis has controlled release and targeting effects, and is currently under experimental research. HA-dissolving MNs of 5-aminolevulinic acid for the treatment of cancer and DHE-dissolving MNs for the treatment of acute migraines are also under experimental research.	[12,63,64,65,66]
Hydrogel-forming MNs	Good biocompatibility Needle mechanical strength and drug delivery rate can be adjusted by changing the density of polymer cross-linking. The drug can be wrapped in the entire MN patch, suitable for high-dose administration. The drug will not be released suddenly, but will pass through the channel continuously at a certain speed, which can prolong the administration time.	Small doses of drugs are easily lost during encapsulation or absorption. Incomplete and uncontrolled drug release.	Hydrogel-forming MNs are widely used in the treatment of arthritis. Melittin-modified HA hydrogel-forming MNs with better effect in the treatment of arthritis can prevent hemolysis and pain caused by injection of purified melittin. The MNs are under experimental research. In addition, hydrogel-forming MNs for the detection of plasma glucose, lactic acid, or chlorine levels, and those for the treatment of non-melanoma skin cancers are also under experimental research.	[18,67,68,69,70]
Stimulus-responsive MNs	Good mechanical properties. Excellent biocompatibility. Effectively improving the specificity of drug delivery and reducing toxicity and side effects.	Poor controllability. Difficult to control the dosage precisely.	The stimuli-responsive MNs currently under experimental research include: hyaluronidase stimuli-responsive MNs for the treatment of tumors and hypoxia-responsive MNs for the treatment of diabetes.	[71,72,73,74]
Bionic MNs	High mechanical strength. Good biocompatibility. Painless insertion.	Complicated machining process and expensive equipment. More difficult to produce.	The invention of bionic MNs plays a role in promoting and inspiring the application of MNs in the fields of biosignal recording, tissue adhesion, and transdermal drug delivery.	[55,75,76,77]

**Table 4 pharmaceutics-14-01736-t004:** Laboratory studies of MN delivery of arthritis drugs in recent years.

Type of MN	Materials	Fabrication Process	Single MN Base Width × Height (μm)	Array Area/cm^2^	Array Number	API	API Classification	Drug Delivery Enhancement Technology	Animal Models	Result	Ref.
Solid MNs	Polycarbonate		×500			Ketoprofen	NSAIDs			When MN and ketoprofen gel were coupled, the AUC and Cmax of ketoprofen dramatically increased and the relative bioavailability was higher.	[127]
	Silicon		×200	0.25	4 × 4	Methotrexate	Conventional synthetic DMARDs			The plasma concentration of methotrexate would increase linearly with increasing number of MNs.	[128]
			×200			Triptolide	Herbal extracts	Liposome hydrogel patch	CIA	The drug delivered by MN could promote transdermal absorption effectively.	[129]
			×250/750			Bee venom	Bio-Drugs		Sodium urate-induced acute gouty inflammation, Lipopolysaccharide (LPS)-induced acute inflammation	MN can promote the percutaneous absorption of the active macromolecules: bee venom gel.	[130]
			100 × 250	0.25		Alkaloids from Aconitum sinomontanum	Herbal extracts	Nanostructured lipid carriers	AIA	MN led to deeper permeation and combination of MN and NLCs; could improve the therapeutic efficacy.	[131]
			×250/500/750/1000			Paeoniflorin	Herbal extracts	Ethosomes		Both ethosome and MN can enhance the penetration of paeoniflorin, and MN shows a more dramatic effect.	[132]
						Paeoniflorin	Herbal extracts	Ethosomes		MN could promote the entry of the ethosomes into the skin and greatly improved the possibility of deep penetration of the water-soluble paeoniflorin.	[133]
Coated MNs	Medical-Grade liquid crystalline polymer	micromolding/solvent casting method	139 ± 17 × 1160 ± 43	1	6 × 6	Lidocaine	Analgesic drug			MNs show faster release of drug than TS and can be used for instant supply of the same drug.	[118]
Hollow MNs		3D printing method				Denosumab	RANKL inhibitor			In comparison to the subcutaneous group, similar rate of release was observed with the 3D printed hollow MN without inducing any stimuli of pain.	[134]
	PVP, PVA	micromolding/spin-casting method	460 × 1200	0.35	9 × 9	Tofacitinib	JAK inhibitor			The amount of drug permeated using MNs is superior to other approaches and dissolving MN shows better ability to promote penetration.	[135]
Dissolving MNs	MT	micromolding/spin-casting method	210 × 700		10 × 10	Methotrexate	Conventional synthetic DMARDs	Iontophoretic delivery		MNA-delivered anti-TNF-α Ab treatment had a therapeutic effect in an animal model of psoriasiform dermatitis and effectively reduced key biomarkers of psoriasiform inflammation including epidermal thickness and IL-1b expression.	[136]
	CMC	drawing lithography	×600		5 × 5/9 × 9	TNF-α antibodies	TNF inhibitor		Imiquimod-induced psoriasiform inflammation	IPS-based DMN-mediated delivery of CAP was able to significantly modulate macrophages for the production of TNF-α, IL-1β, and IL-6 compared to topical application.	[137]
	HA, PVP	micromolding method	380 × 680		13 × 13	Capsaicin	Lipophilic drugs	Innovative polymeric system	CIA	DMNs resulted in lower peak plasma levels but higher plasma ARM concentration at 8 h after administration and could reverse paw edema, similar to ARM intramuscular injection.	[138]
	Oligo-HA	micromolding/solvent casting method	300 × 800		7 × 10	Artemether	Lipophilic drugs		CIA	SH-DM significantly enhanced the permeation rate of drug compared to the control of SH-G and AUC, and RBA value of SH-DM was 1.99 times higher than that of SH-G.	[139]
	MT, PLGA	micromolding/spin-casting method	×1500		35	Sinomenine	Herbal extracts			The MN patch showed a significant drug deposition within skin (63.37%) and an improved transdermal flux (1.60 μg/cm^2^/h) with a 2.58-fold enhancement in permeation compared to plain drug solution.	[140]
	PVP, PVA	micromolding method	55.42 ± 8.66 × 508.46 ± 9.32		28	Meloxicam	NSAIDs		Carrageenan-induced arthritis	A synergistic 25-fold enhancement of delivery was observed in vivo when a combination of MNs and iontophoresis was used compared with either modality alone.	[141]
	Acrylate-modified HA	micromolding/spin-casting method	300 × 800		15 × 15	Etanercept	TNF inhibitor		AIA	MN showed good bioequivalence to the classical subcutaneous injection administration.	[142]
	PVP, CS, CMC	micromolding/spin-casting method	300 × 500		12 × 12	Neurotoxin	Analgesic drug		CIA	DMNs-NT showed favorable biocompatibility and the skin penetration depth and the cumulative of NT in DMNs-NT was much higher than the NT solution.	[143]
	PVP	micromolding method				Methotrexate	Conventional synthetic DMARDs	Multiple emulsion (w/o/w type) system	AIA	The MN patch significantly suppressed paw swelling compared to positive control.	[64]
	PVP	micromolding method	300 × 350			aconitine	Herbal extracts	Nanostructured lipid carriers	AIA	DMNs showed a higher AUC by enhancing the transdermal delivery efficiency of the ACO-NLCs.	[144]
	PVP-K30	micromolding/vacuum method	300 × 550	1	20 × 20	polydatin	Herbal extracts	Hydroxypropyl-β-cyclodextrin inclusion complexes	Monosodium urate-induced acute gouty arthritis	The complex-loaded DMNs showed better therapeutic effects on the arthritic mice and lower toxicity.	[145]
	HA, Methacrylate-modified HA	micromolding/two-step filling method	×700	0.81	10 × 10	Melittin	Bio-Drugs		AIA	HA-based MN could be as effective as SC injection in inhibiting the progression of RA, and simply modified HAMN with cross-linkable groups showed slow-release properties.	[67]
	PVP	micromolding/vacuum method	200 × 650		12 × 12	Methotrexate	Conventional synthetic DMARDs		AIA	The drug-loaded MN treatment showed better and faster therapeutics compared with the oral groups because of the avoidance of the first-pass effect and sustained release effect.	[146]
	PVP K30, CS, PVA	micromolding/spin-casting method	300 × 600			Brucine	Herbal extracts		AIA	Bru-MN indicated an effective role in inhibiting toe swelling in RA rats, achieving the same effects as methotrexate.	[147]
	HA, PVA	micromolding/spin-casting method	350 × 800		11 × 11	Triptolide	Lipophilic drugs	Liposome	Monosodium iodoacetate-induced osteoarthritis	TP-Lipo@DMNs had a slow-release effect compared with intra-articular injection and significantly reduced knee joint swelling and the level of inflammatory cytokines.	[94]
	HA, Dextran, PVP K17	micromolding/spin-casting method	200 × 600		12 × 12	Tacrolimus, Diclofenac	Calcineurin inhibitor, NSAIDs		Carrageenan/kaolin-induced arthritis	The layered MNs had stronger effects on inhibiting disease development than the other MN groups and injection groups.	[148]
	HA, PVA, Polysaccharides	micromolding/vacuum method	600 × 500		15 × 15	Tetrandrine	Herbal extracts	Calcium carbonate-hybridized PLGA nanocarrier	AIA	Tet-6 s-NP (CaCO3)/GP-MN strongly reduced synovial inflammation and angiogenesis, exerting a most obvious anti-inflammatory effect on rats with AA.	[149]
	PVP/VA	micromolding/vacuum method	260 × 504	1	20 × 20	Indomethacin	NSAIDs	Mixed micelles		Mixed micelle-loaded DMNs showed much shorter lag time and higher bioavailability compared to the commercial patch.	[150]
Hydrogel-forming MNs	PVA, MT	micromolding/freezing and thawing method	×500			Sinomenine hydrochloride	Herbal extracts			The sinomenine hydrochloride (SH) in SH-loaded MT/ PVA MN exhibited lower clearance, longer retention time, higher bioavailability and stability versus SH-loaded hydrogel.	[151]
	PVA	micromolding method	300 × 729.5 ± 11.2		11 × 11	Methotrexate	Conventional synthetic DMARDs			PVA-based HFMNs delivered variable doses of drug through skin more efficiently compared with the previous HF-MNs and could be removed without leaving any measurable residues.	[152]
	PVA	micromolding/solvent casting method	300 × 729.5 ± 50		11 × 11	Methotrexate	Conventional synthetic DMARDs			The HFMN patch was able to deliver MTX (around 40% of the applied dose) in a controlled and sustained manner.	[153]
	Methacrylate-modified HA	micromolding/vacuum method	300 × 800	1	15 × 15	DTA6	DEK protein inhibitors		CIA	HMN had similar or better efficacy than intravenous injection and would efficiently alleviate arthritis and profoundly improve the compliance of patients.	[115]
	PVA, PVP K90, HPMC, PEG4000, PEG10000, Glycerol	micromolding/spin-casting method	300 × 800	0.5	11 × 11	Methotrexate	Conventional synthetic DMARDs			HFMN could deliver MTX in a sustained manner over 24 h, with significantly lower Cmax, while maintaining the same or even better delivered dose than that achieved by the oral administration route.	[126]

CMC: Carboxymethyl Cellulose; CS: Chondroitin sulfate; HPMC: Hydroxy Propyl Methyl Cellulose; HA: Hyaluronic Acid; MT: Maltose; PVP: Polyvinylpyrrolidone; PVA: Polyvinyl Alcohol; PEG: Polyethylene Glycol; PLGA: Poly(lactic-co-glycolic Acid); CIA: Collagen-induced Arthritis; AIA: Adjuvant-induced arthritis.

**Table 5 pharmaceutics-14-01736-t005:** Current clinical trials of microneedle delivery of arthritis drugs.

	NCT number	Title	Status	Interventions	Population	Date	Locations
1	NCT03607903	Adalimumab Microneedles in Healthy Volunteers	Phase 1\2 Completed	Adalimumab ID (microneedle: MicronJet600)\SC	Enrollment: 24 Age: 18 to 45 years Sex: All	11 July 2018 to 30 October 2018	Centre for Human Drug Research, Leiden, Netherlands

ID: intradermal.

## Data Availability

Not applicable.

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
