# Peer review of "Promising Strategies for Transdermal Delivery of Arthritis Drugs: Microneedle Systems"

_pharmaceutics, 2022, doi:10.3390/pharmaceutics14081736_

Round 1

Reviewer 1 Report

Overall comments:

In the article authors have reviewed development and use of microneedle technology, for transdermal drug delivery for arthritis treatment. Authors have done extensive review of the literature discussing different types of microneedle technologies employed for delivering various drugs either transdermally or to synovial cavity. Providing an overview on the state of the art of the field. Researchers working in the field of microneedle technology development or on arthritis treatment will find this work useful for their studies.

However, I think minor adjustments need to be made before consideration for the publication.

Comments:

1.       Section 2. Type of arthritis and Page 3, line 138-140.

readers will find it easy to follow and refer to if the information on the different type of arthritis, patient age group, and underlying cause is also provided in a tabular format.

2.       Page 4, table 1: OTC products, Turmeric

I think OTC products is not a category in itself. Several other dugs mentioned in the table can fall under this (OTC) category. Turmeric has some active pharmaceutical constituents which may be responsible for its anti-inflammatory activity, they should be mentioned instead. Kindly correct.

3.       Page 6, line 192, “oral drugs delivery via MNs”, this line is confusing. If a drug is delivered orally then it is oral drug, if it is delivered via transdermal route, it is a transdermal drug. Kindly correct.

4.       Page 6, line 196-197, “Some biological drugs…...through injection”. Kindly check for the correctness of the sentence.

5.       Page 11, line 304-305, “The faster the…….faster the drug takes effect”. This statement is not technically correct. Dissolution of coating can’t be directly correlated with response of drug, it can only be correlated with release of the drug from a dosage form.

6.       Table 3.

It will be highly useful for the readers of the dimensions (length x breadth) and fabrication process used are also tabulated here, along with other information already provided.

7.       Page 20 line 631, using term “drug development” is technically incorrect here, kindly use “dosage form development”.

8.       Page 20, line 644, most of the MNs are developed for OTC application, if OCT procedure is employed for application of MNs, it fails to be an OTC product. As visit to specialized facility is must in this scenario. As mentioned by authors on page 22 line 718 “MNs can be applied by patients without needing to visit the clinic. Kindly explain or correct accordingly.

9.       Page 21, section 7. Kindly provide information on the typical usage/implantation duration for an MN post application. How frequently they will require a replacement.

10.   Page 17, line 473. Use “These” instead of “This”.

11.   Page 18, line 516, remove “and”

12.   Page 20, line 652, correct for “bioavailablility”.

13.   Page 21, line 704, kindly replace “the” with “at”.

14.   Kindly correct “conflicts of interest” statement.

Author Response

Response to Reviewer 1 Comments

Thank you very much for your careful review. We have carefully revised the article after receiving the amendment opinions. The following is the reply to your amendment opinions. For details, please refer to document.

Point 1: Section 2. Type of arthritis and Page 3, line 138-140.

readers will find it easy to follow and refer to if the information on the different type of arthritis, patient age group, and underlying cause is also provided in a tabular format.

Response 1: We have added a new table (Table 1) to intuitively display the contents of various arthritis.

Point 2: Page 4, table 1: OTC products, Turmeric

I think OTC products is not a category in itself. Several other dugs mentioned in the table can fall under this (OTC) category. Turmeric has some active pharmaceutical constituents which may be responsible for its anti-inflammatory activity, they should be mentioned instead. Kindly correct.

Response 2: We replaced “Turmeric” with its main component “Curcumin” in Table 2, and classified it as a supplement according to the existing literature (doi: 10.2147/DDDT.S117432; 10.1177/2156587216636747). Because curcumin has a wide range of pharmacological effects, it cannot be classified as an analgesic or anti-inflammatory drug, and herbal medicine and its extracts are often classified as dietary supplements, so we classify curcumin as a supplement.

Point 3: Page 6, line 192, “oral drugs delivery via MNs”, this line is confusing. If a drug is delivered orally then it is oral drug, if it is delivered via transdermal route, it is a transdermal drug. Kindly correct.

Response 3: We corrected the statement of this sentence (Page 6, line 192) and deleted the word "oral" to avoid ambiguity.

Point 4: Page 6, line 196-197, “Some biological drugs…...through injection”. Kindly check for the correctness of the sentence.

Response 4: We have corrected the statement of this sentence (Page 6, line 196-197) to make it accurate.

Point 5: Page 11, line 304-305, “The faster the……faster the drug takes effect”. This statement is not technically correct. Dissolution of coating can’t be directly correlated with response of drug, it can only be correlated with release of the drug from a dosage form.

Response 5: We corrected the sentence (Page 11, line 304-305) "the faster the drug takes effect" to "the faster the drug releases”.

Point 6: Table 3.

It will be highly useful for the readers of the dimensions (length x breadth) and fabrication process used are also tabulated here, along with other information already provided.

Response 6: As a table is added before Table 3, it becomes Table 4. And we have added several columns of information in Table 4 to fully show the MNs.

Point 7: Page 20 line 631, using term “drug development” is technically incorrect here, kindly use “dosage form development”.

Response 7: Page 20, line 642.We corrected "drug development" to "dosage form development".

Point 8: Page 20, line 644, most of the MNs are developed for OTC application, if OCT procedure is employed for application of MNs, it fails to be an OTC product. As visit to specialized facility is must in this scenario. As mentioned by authors on page 22 line 718 “MNs can be applied by patients without needing to visit the clinic. Kindly explain or correct accordingly.

Response 8: OCT program is only used as an experimental instrument in the development and safety evaluation of microneedles. OCT is not involved in the application of MNs to patients. In addition, compared to solid metal microneedles and hollow MNs, the matrix of dissolving MNs is soluble and biocompatible, so it does not need to be retrieved even if the needle breaks in the body. Therefore, dissolving MNs does not need to go to the hospital to rely on professional operation, and patients can apply them on their own. Therefore, dissolving MNs still have the potential to become OTC products.

Point 9: Page 21, section 7. Kindly provide information on the typical usage/implantation duration for an MN post application. How frequently they will require a replacement.

Response 9: Because of the variety of microneedles and the wide selection of matrix, the use duration of microneedles is mainly affected by the design. At present, there are few clinical trials of microneedles, and it is difficult to provide specific duration and replacement time.

Point 10-14:

  1. Page 17, line 473. Use “These” instead of “This”.
  2. Page 18, line 516, remove “and”
  3. Page 20, line 652, correct for “bioavailablility”.
  4. Page 21, line 704, kindly replace “the” with “at”.
  5. Kindly correct “conflicts of interest” statement.

Response 10-14: It has been revised according to your modification opinions.

Reviewer 2 Report

The application of microneedles to the skin surface creates micron-sized transport routes that enable faster drug delivery, bypassing the first-pass effect, and alleviating patient discomfort associated with injections. The purpose of this review article is to focus mainly on the use of microneedles (MNs) in the treatment of arthritis. This review summarizes the different types of arthritis and the currently used therapies. Different types of MNs such as solid, hollow, coated, dissolving, bionic, hydrogel-forming and stimulus-responsive MNs and their use in drug delivery for the treatment of various types of arthritis are discussed. The Authors drew attention to the insufficient number of clinical trials confirming their effectiveness and safety.

In my opinion, the manuscript is ready to be published after a minor revision. Further comments about the manuscript are reported below.

1. Literature references should be written in accordance with the journal's requirements: (Author 1, A.B.; Author 2, C.D. Title of the article. Abbreviated Journal Name Year, Volume, page range.).

2. In Table 3, the column entitled 'Years of Publication' can be removed and replaced by a column entitled, for example, 'Results' or 'Advantages'. To facilitate editing, the font can be reduced.

3. The text uses different line spacing (e.g. lines 348-369, 422-426).

Author Response

Response to Reviewer 2 Comments

Thank you very much for your careful review. We have carefully revised the article after receiving the amendment opinions. The following is the reply to your amendment opinions. For details, please refer to document.

Point 1: Literature references should be written in accordance with the journal's requirements: (Author 1, A.B.; Author 2, C.D. Title of the article. Abbreviated Journal Name Year, Volume, page range.).

Response 1: We have reviewed the references in detail and corrected the errors therein.

Point 2: In Table 3, the column entitled 'Years of Publication' can be removed and replaced by a column entitled, for example, 'Results' or 'Advantages'. To facilitate editing, the font can be reduced.

Response 2: As a table is added before Table 3, it becomes Table 4. And we changed the "publication years" in Table 4 to "results".

Point 3: The text uses different line spacing (e.g. lines 348-369, 422-426).

Response 3: We corrected different wrong line spacing.

Reviewer 3 Report

This manuscript aims to highlight the developments in the delivery of drugs using microneedles in the treatment of arthritis. Furthermore, the challenges in the translation of microneedle-based delivery from the laboratory to clinical practice were discussed. The manuscript reads well and the authors have nicely summarized the developments in the concerned area. The authors could discuss some of the ideal properties of drug molecules that make them suitable for this mode of delivery. Adding a summary of ongoing/completed clinical trials related to the use of microneedles in arthritis treatment, if any, would be beneficial. 

Author Response

Response to Reviewer 3 Comments

Thank you very much for your careful review. We have carefully revised the article after receiving the amendment opinions. The following is the reply to your amendment opinions. For details, please refer to document.

Point 1: The authors could discuss some of the ideal properties of drug molecules that make them suitable for this mode of delivery.

Response 1: We added a discussion on the molecular properties of ideal drugs suitable for microneedle administration in Part 7 - Conclusion and future perspectives (Page 26, line 723-726).

Point 2: Adding a summary of ongoing/completed clinical trials related to the use of microneedles in arthritis treatment, if any, would be beneficial.

Response 2: We added a summary table (Table 4) on clinical trials of using MNs to treat arthritis. Although it has only one piece of information, it is still helpful.
